# Effectiveness of BNT162b2 and mRNA-1273 Vaccines against COVID-19 Infection: A Meta-Analysis of Test-Negative Design Studies

**DOI:** 10.3390/vaccines10030469

**Published:** 2022-03-18

**Authors:** Shuailei Chang, Hongbo Liu, Jian Wu, Wenwei Xiao, Sijia Chen, Shaofu Qiu, Guangcai Duan, Hongbin Song, Rongguang Zhang

**Affiliations:** 1Department of Epidemiology, College of Public Health, Zhengzhou University, Zhengzhou 450001, China; changshuailei97@126.com (S.C.); wujianzzu@163.com (J.W.); 18463723394@163.com (W.X.); gcduan@zzu.edu.cn (G.D.); 2Chinese PLA Center for Disease Control and Prevention, Beijing 100071, China; mailoflhb@126.com (H.L.); qiushf0613@hotmail.com (S.Q.); 3Department of Epidemiology, College of Public Health, China Medical University, Shenyang 110122, China; 2020120258@stu.cum.edu.cn; 4Department of Epidemiology, College of Public Health, Hainan Medical University, Haikou 571199, China

**Keywords:** meta-analysis, test-negative design, vaccine effectiveness, BNT162b2 vaccine, mRNA-1273 vaccine, COVID-19 infection

## Abstract

Although numerous COVID-19 vaccines are effective against COVID-19 infection and variants of concern (VOC) in the real world, it is imperative to obtain evidence of the corresponding vaccine effectiveness (VE). This study estimates the real-world effectiveness of the BNT162b2 and mRNA-1273 vaccines against COVID-19 infection and determines the influence of different virus variants on VE by using test-negative design (TND) studies. We systematically searched for published articles on the efficacy of BNT162b2 and mRNA-1273 against COVID-19 infection. Two researchers independently selected and extracted data from eligible studies. We calculated the VE associated with different vaccine types, SARS-CoV-2 variants, and vaccination statuses, using an inverse variance random-effects model. We selected 19 eligible studies in the meta-analysis from 1651 records. For the partially vaccinated group, the VE of BNT162b2 and mRNA-1273 was 61% and 78% against COVID-19 infection, respectively. For the completely vaccinated group, the VE of BNT162b2 and mRNA-1273 was 90% and 92% against COVID-19 infection, respectively. During subgroup analyses, the overall VE of BNT162b2 and mRNA-1273 against the Delta variant was 53% and 71%, respectively, for the partially vaccinated group; the respective VE values were 85% and 91% for the fully vaccinated group. Irrespective of the BNT162b2 or mRNA-1273 vaccines, the Delta variant significantly weakened vaccine protection for the partially vaccinated group, while full vaccination was highly effective against COVID-19 infection and various VOC. The mRNA-1273 vaccine is more effective against COVID-19 infection and VOC than the BNT162b2 vaccine, especially for the partially vaccinated group. Overall, the results provide recommendations for national and regional vaccine policies.

## 1. Introduction

The global spread of severe acute respiratory syndrome coronavirus 2 (SARS-CoV-2) has resulted in more than 255 million confirmed cases and more than 5 million deaths as of 19 November 2021. In addition, the global COVID-19 situation remains severe, with the number of new novel coronavirus cases going up 0.5 million per day. The advent of COVID-19 vaccines and widespread vaccination across the globe have offered glimmers of hope for the eradication of COVID-19, and more than 7.3 billion dose of vaccine have been administered as of 17 November 2021 [1]. Both BNT162b2 and mRNA-1273 vaccines have been shown to be highly effective in multiple randomized clinical trials (RCTS) [2,3,4,5,6].

However, clinical trials may not simulate the real-world mass vaccination of COVID-19 vaccines because these studies were conducted in highly controlled environments. In contrast, the emerging variants defined as variants of concern (VOC), including the Alpha, Beta, Gamma, and Delta variants, were associated with increased transmission, more severe outcomes, reduced antibody neutralization from previous infection or vaccination, and lower vaccine effectiveness [7,8,9,10]. Thus, it is urgent and necessary to synthesize evidence of the vaccine effectiveness (VE) of COVID-19 vaccines against COVID-19 infection and VOC in the real world.

Because of feasibility and ethical issues, observational studies will replace randomized controlled trials as the preferred method of assessing the efficacy of COVID-19 vaccines. Test-negative design—one of the main methods for assessing the effectiveness of influenza vaccines—has increasingly been used by researchers as a convenient and effective way to assess the efficacy of COVID-19 because it reduces the bias due to infection misclassification and confusion in care-seeking behavior compared to other observational studies [11,12,13,14]. More importantly, when a large randomized controlled trial is impossible, the design can detect changes in vaccine effectiveness or measure VE against VOC [15].

Global mass vaccination requires a synthesis of evidence on SARS-CoV-2 VE to drive global health policy strategies. Especially, the VE against VOC may influence the choices of policy makers with regard to vaccine supply, procurement, deployment, and vaccination. We conducted a meta-analysis using TND studies to assess the VE of BNT162b2 and mRNA-1273 vaccines against COVID-19 infection and VOC.

## 2. Materials and Methods 

### 2.1. Data Sources and Search Strategies

We systematically searched for articles published from 2020, when the first vaccine against COVID-19 was released, to October 26, 2021, in the following electronic databases: PubMed, Cochrane Library, Web of Science, Embase, Scopus, and preprint servers (Biorxiv and Medrxiv). We employed literature management with EndNote-X9. This meta-analysis used the following combinations as search terms: “SARS-CoV-2, COVID-19, 2019nCoV, Pfizer, BNT162b2, Comirnaty, mRNA-1273, Spikevax, Moderna, mRNA vaccine, effectiveness, efficacy, test-negative case-control, test-negative design.” Terms could appear in either the title or abstract of the article. There were no restrictions on the language of the articles that were reviewed, and the reference lists of related articles were also browsed to retrieve additional studies. 

Results were screened independently by two reviewers, and disputes were resolved in consultation with a third reviewer. Only TND studies evaluating the effectiveness of BNT162b2 or mRNA-1273 vaccine against SARS-CoV-2 infection were included. Studies eligible for inclusion should meet the following criteria: definition of case is people testing positive for SARS-CoV-2 confirmed by PCR and control is people testing negative for SARS-CoV-2 confirmed by PCR [14]; the efficacies of BNT162b2 and mRNA-1273 vaccines were reported separately, and measurements used to assess the VE were adjusted. This meta-analysis excluded studies if they contained exclusively mRNA vaccine VE data (or mixed BNT162b2 and mRNA-1273 VE data which could not be separated). Studies where the outcome was not COVID-19 infection, such as hospitalization, severe illness, death, and other severe outcomes, were excluded. 

### 2.2. Data Extraction and Study Quality Assessment

Two research personnel extracted data independently from selected studies, and a third author was consulted when consensus was lacking. The following data from selected articles were extracted: first author’s surname, country, sample, total number of participants, number of vaccinated and unvaccinated individuals in the cases and controls, type of vaccine, variants of COVID-19, adjusted VE estimate, and adjustment factors. We only extracted data related to SARS-CoV-2 infection when both the results of SARS-CoV-2 infection and symptomatic infection were reported. Quality assessment of researches was conducted with the Newcastle–Ottawa scale (NOS) [16] by a researcher independently, and the second researcher was responsible for inspection and verification. The score >7 was considered high quality, 5–6 was medium, <5 was low, and a study with a score of 5 or greater was included in the meta-analysis. 

### 2.3. Data Synthesis and Analysis

Relevant study characteristics were synthesized in a tabular format. VE against SARS-CoV-2 infection and VOC with different vaccination situations were evaluated with the odds rate (OR) of vaccination between cases and controls and were calculated using the following formula: VE = (1 − OR) × 100%. One-dose vaccination and two-dose vaccination were defined as partial and full vaccination, respectively. Meta-analysis with the random-effects model was employed to calculate the pooled OR and 95% confidence intervals, and statistical heterogeneity between the studies was assessed using *I*^2^. We performed subgroup analysis stratifying by different times after vaccination, VOC, and vaccine situation. All analyses were conducted using Stata 11.2 (StataCorp, College Station, TX, USA). A two-tailed *p* value < 0.05 was considered to be statistically significant.

## 3. Results

### 3.1. Literature Research and Study Characteristics

The flow diagram of the literature screening in this research is shown in Figure 1. We identified 1681 publications or preprints, after excluding duplicates and irrelevant literature, 52 documents were reviewed in full text for eligibility, and the remaining 19 articles [17,18,19,20,21,22,23,24,25,26,27,28,29,30,31,32,33,34,35] were included in the present meta-analysis. 

The study characteristics of the selected articles are shown in Table 1, Table 2 and Table 3 (more details are provided in Table A1 andTable A2). In total, six articles were produced in Canada [18,19,22,30,31,32], five in the United States (US) [21,23,24,25,26], four in the United Kingdom (UK) [20,29,34], three in Qatar [17,28,35], and one from multiple-centers from UK, France, Ireland, the Netherlands, Portugal, Scotland, Spain, and Sweden [33] among the eligible research. The numbers of documents that reported the VE of BNT162b2 and mRNA-1273 against infection was 11, only mRNA-1273, 2, and only BNT162b2, 6. A total of 12 articles [17,18,20,22,23,24,27,28,29,32,34,35] reported the VE against VOC (Alpha, Beta, Gamma, and Delta), of which five mentioned only the BNT162b2 vaccine [20,27,28,29,35], one only the mRNA-1273 vaccine [24], and six compared the BNT162b2 and mRNA-1273 vaccines [17,18,22,23,32,35]. The quality assessment for the included studies was implemented by NOS, and all research had scores greater than 5, meeting the criteria for inclusion in the meta-analysis.

### 3.2. Pooled VE of BNT162b2 and mRNA-1273 against COVID-19 Infection

In the meta-analyses of VE against COVID-19 infection, we grouped the studies by four different vaccination situations: BNT162b2, partially vaccinated; BNT162b2, fully vaccinated; mRNA-1273, partially vaccinated and mRNA-1273, fully vaccinated. The meta-analysis of twelve articles that used OR as VE evaluation indicated the prominent role against confirmed COVID-19 infection by RT-PCR after partial vaccination of BNT162b2 vaccine (pooled OR = 0.39; 95% CI: 0.33–0.45, *I*^2^ = 96.7%), where the pooled estimate showed a VE of 61% (95% CI: 55–67%) (Figure 2). Additionally, the meta-analysis of eight articles which used OR as the VE evaluation indicated the prominent role against confirmed COVID-19 infection by RT-PCR after partial vaccination with the mRNA-1273 vaccine (pooled OR = 0.22; 95% CI: 0.17–0.27, *I*^2^ = 88.1%), where the pooled estimate showed a VE of 78% (95% CI: 73–83%) (Figure 2).

The vaccine effectiveness was further improved after the subjects had been fully vaccinated. Higher vaccine effectiveness was observed after full vaccination with the BNT162b2 vaccine, where the meta-analysis of 15 articles which used OR reported a pooled OR of 0.10 (95% CI: 0.09–0.12; *I*^2^ = 98.8%) and a VE of 90% (95% CI: 88–91%) (Figure 2). Similarly, the phenomenon of higher vaccine effectiveness was also observed after full vaccination with the mRNA-1273 vaccine, where the meta-analysis of 10 articles which used OR reported a pooled OR of 0.08 (95% CI: 0.06–0.09; *I*^2^ = 94.7%) and VE of 92% (95% CI: 91–94%) (Figure 2).

### 3.3. Pooled VE against VOC

To assess the vaccine effectiveness of BNT162b2 and mRNA-1273 vaccines against VOC, we performed a subgroup analysis for VOC in partially vaccinated and fully vaccinated individuals of BNT162b2 and mRNA-1273 vaccine. In the partial vaccination groups, the summary VE of the BNT162b2 vaccine against the Delta, Alpha, Beta and Gamma variants was 53% (OR = 0.47, 95% CI: 0.36–0.61), 37% (OR = 0.37, 95% CI: 0.27–0.52), 36% (OR = 0.64, 95% CI: 0.44–0.93) and 72% (OR = 0.28, 95% CI: 0.16–0.49), respectively, and the results of the pooled estimate showed that the VE of the mRNA-1273 vaccine against Delta, Alpha and Gamma variants was 71% (OR = 0.29, 95% CI:0.24–0.35), 85% (OR = 0.15, 95% CI: 0.11–0.21) and 87% (OR = 0.13, 95% CI: 0.08–0.22) (Figure 3), respectively. In the full vaccination groups, the total VE of BNT162b2 against Delta, Alpha and Gamma variants was 85% (OR = 0.15, 95% CI: 0.11–0.20), 94% (OR = 0.06, 95% CI: 0.04–0.10) and 92% (OR = 0.08, 95% CI: 0.05–0.14), respectively (Figure 3), whereas the pooled VE of the mRNA-1273 vaccine against Delta, Alpha and Gamma was 91% (OR = 0.09, 95% CI: 0.06–0.13), 95% (OR = 0.05, 95% CI: 0.02–0.12) and 93% (OR = 0.07, 95% CI: 0.04–0.13), respectively (Figure 3), respectively.

### 3.4. Pooled VE by Different Time Points after Vaccination

In the subgroup analysis of VE by different time points after vaccination, we conducted studies on the following groups: 14 or more days and 21 or more days after partial vaccination with BNT162b2 vaccine and mRNA-1273 vaccine and 7 or more days and 14 or more days after full vaccination with BNT162b2 vaccine and mRNA-1273 vaccine. The results indicated that the summary VE of 14 or more days after partial vaccination with BNT162b2 vaccine was 58% (OR = 0.42, 95% CI: 0.35–0.50), while that of the mRNA-1273 was 77% (OR = 0.23, 95% CI: 0.17–0.30) (Figure 4). Of the people with partial vaccination with BNT162b2 or mRNA-1273 after 21 or more days, the pooled VE was 67% (OR = 0.33, 95% CI: 0.23–0.47) and 80% (OR = 0.20, 95% CI: 0.15–0.26) (Figure 4), respectively. Studying the fully vaccinated individuals showed that the total VE of full vaccination with BNT162b2 or mRNA-1273 after 7 or more days was 89% (OR = 0.11, 95% CI: 0.10–0.13) and 94% (OR = 0.06, 95% CI: 0.04–0.08), and that after 14 or more days was 90% (OR = 0.10, 95% CI: 0.08–0.13) and 92% (OR = 0.08, 95% CI: 0.06–0.10) (Figure 4), respectively.

## 4. Discussion

In this meta-analysis of the VE against COVID-19 infection and VOC from TND studies, in both BNT162b2 and mRNA-1273, partial vaccination had low VE against COVID-19 infection with VE of 61% and 78%, respectively, and full vaccination had significantly higher efficacy against COVID-19 infection than the partial vaccine with VE of 90% and 92%, respectively, which was consistent with, but slightly less effective than the phase 3 randomized controlled trials of BNT162b2 [5] and mRNA-1273 [4] vaccines. The differences in vaccine effectiveness between clinical trials and real-world studies may result from differences in the definition of COVID-19. In clinical trials, the presence of symptoms and a positive SARS-CoV-2 RT-PCR test was defined as confirmed COVID-19, whereas in the studies included in our meta-analyses, only a positive SARS-CoV-2 RT-PCR test confirmed COVID-19. In addition, some studies have reported that vaccines are less effective in protecting older adults against COVID-19 infection, and four articles (Skowronski et al. [18], Thompson et al. [21], Bernal et al. [27] and Kissling et al. [33]) included in our analysis studied vaccine effectiveness in adults over the age of 50, which could reduce the overall vaccine protection. Furthermore, a low sensitivity or specificity of PCR tests in TND may lead to misclassification of cases and controls, which would also weaken estimates of vaccine effectiveness [20].

Similar to the results of the vaccine for preventing infection, full vaccination was significantly more effective for preventing hospitalization than partial vaccination. The difference is that the effectiveness of vaccines in preventing hospitalization is slightly higher than that of preventing infection under the same vaccine status (Table A3, Figure A1). Our findings also showed that partial vaccination of both BNT162b2 and mRNA-1273 vaccines significantly reduced the effect on Delta compared with the Alpha variant, but the difference was significantly reduced after full vaccination. The same phenomenon was observed in two U.S. CDC reports [36,37] and an observational study on the efficacy of vaccines against Delta infection [38]. In this analysis, the estimated vaccine effectiveness of partial BNT162b2 and mRNA-1273 vaccines for the Delta variant was 53% and 71%, respectively, and that of full vaccination for the Delta variant was 85% and 91%, respectively. In the subgroup analysis, the VE of mRNA-1273 was slightly higher at 7 or more days after full vaccination than at 14 or more days, which we believed was caused by the significantly higher proportion of studies on Delta in the group of 14 or more days than in the group of 7 or more days.

The Delta variant, first identified in India in December 2020 [39,40], quickly became the dominant reported variant in the country in just a few months and is now the most common variant of COVID-19 infections globally, making universal vaccination undoubtedly the best protection against COVID-19 infection in the current situation. The included studies [28,35] showed that partial vaccination had the lowest efficacy against the Beta variants, with BNT162b2 and mRNA-1273 vaccines having efficacies of only 36% and 61%, respectively. However, the number of studies on vaccine efficacy against the Beta variants included in this analysis was too small (less than 3) to provide valuable evidence of vaccine efficacy against Beta as well as Gamma variants.

We also found that mRNA-1273 seemed to be better than BNT162b2 against COVID-19 infection and the VOC after both partial and full vaccination, which is consistent with the results of several other articles [41,42,43]. In particular, when partial vaccination was given, the efficacy of mRNA-1273 against COVID-19 infection (78%) was 17 percentage points higher than that of BNT162b2 (61%). The efficacy of mRNA-1273 against Delta (71%) was 18 percentage points higher than that of BNT162b2 (53%). Differences in VE between the mRNA-1273 and BNT162b2 vaccines might be due to higher mRNA content in mRNA-1273 compared with BNT162b2 and the longer interval between priming and boosting for mRNA-12733 (4 weeks vs. 3 weeks for BNT162b2) [44,45]. Self W.H. also indicated that the VE of mRNA-1273 (VE = 93%) against COVID-19 hospitalization was higher than that of the BNT162b2 vaccine (VE = 88%) (*p* = 0.011) [46]. We believe that prioritizing the mrNA-1273 vaccine may be a better option in countries and regions where vaccine stocks are insufficient for full vaccination.

There are some limitations in our meta-analysis because TDN studies lack a detailed protocol for consensus, in terms of experimental design. There were six articles involving individuals with COVID-19 symptoms, two articles involving health care workers, and other articles involving individuals or community residents undergoing tests among the 19 articles analyzed, and samples from different sources might have skewed the VE of the vaccines. Each study was not exactly the same in terms of adjustment factors; however, key factors such as age and gender were adjusted for all the literature. In addition, nine articles did not report the strains of COVID-19, and almost all the research was conducted in four countries (US, UK, Canada, and Qatar). These differences may have contributed to the heterogeneity observed in some of the pooled analyses. Due to inconsistent grouping of age, risk factors, and severe disease events in the included studies, we did not extract sufficient data for subgroup analysis of these factors. This analysis also does not analyze the efficacy of long periods after vaccination and protection efficiency against severe illness and death in hospital. Despite the limitations described above, our analyses contribute significantly to the evidence base and highlights the spots that may exist in VE across type of vaccine, status of vaccination, and against VOC.

## 5. Conclusions

Irrespective of the BNT162b2 or mRNA-1273 vaccine, the Delta variant significantly weakened vaccine protection after partial vaccination, and full vaccination is highly effective against COVID-19 infection and the VOC. The mRNA-1273 vaccine is more effective against COVID-19 infection and VOC than the BNT162b2 vaccine, both after partial and full vaccination. The evidence in this study provides recommendations for national and regional vaccine policies.

## Figures and Tables

**Figure 1 vaccines-10-00469-f001:**
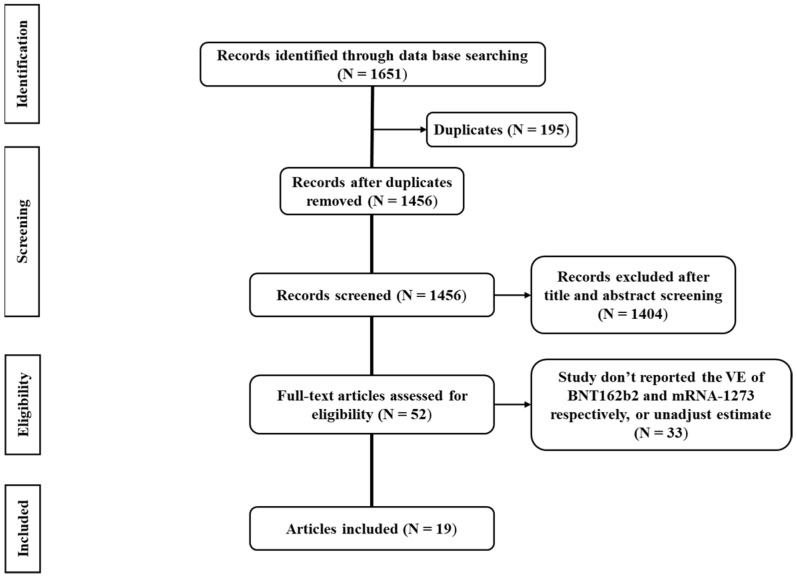
Preferred Reporting Items for Systematic Reviews and Meta-analyses (PRISMA) flow diagram of the process of study selection.

**Figure 2 vaccines-10-00469-f002:**
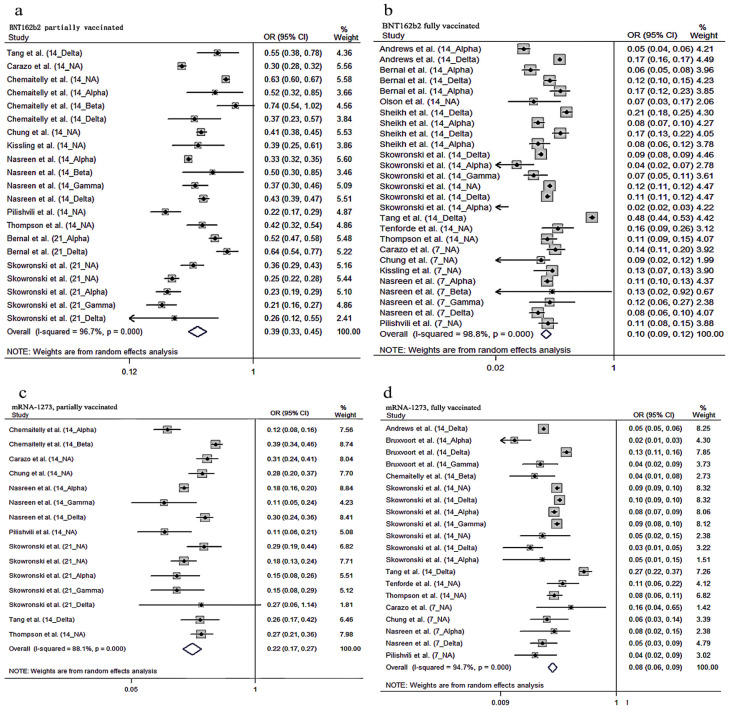
Forest plot odds ratio (OR) against COVID-19 infection (**a**): OR with 95% confidence intervals (CI) against COVID-19 infection (partially vaccinated BNT162b2); (**b**): OR with 95% confidence intervals (CI) against COVID-19 infection (fully vaccinated BNT162b2); (**c**): OR with 95% confidence intervals (CI) against COVID-19 infection (partially vaccinated mRNA-1273); (**d**): OR with 95% confidence intervals (CI) against COVID-19 infection (fully vaccinated mRNA-1273).

**Figure 3 vaccines-10-00469-f003:**
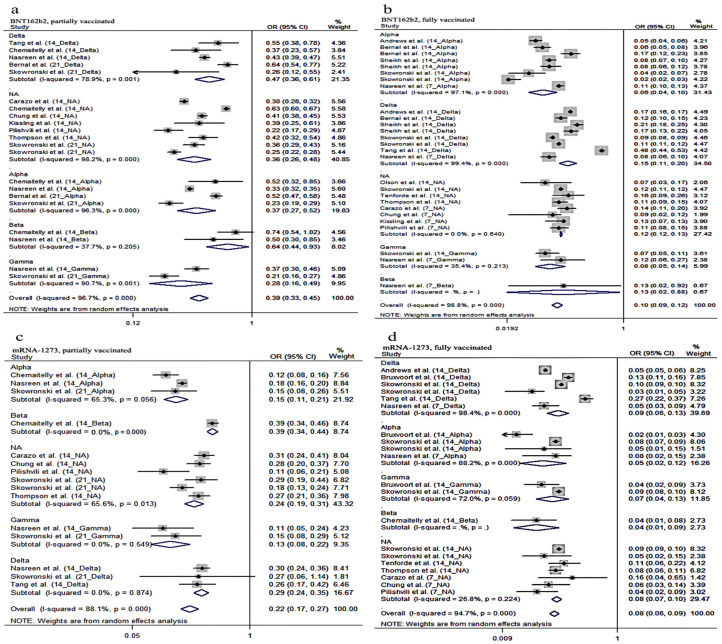
Forest plot odds ratio (OR) against VOC (**a**): OR with 95% confidence intervals (CI) against VOC (partially vaccinated BNT162b2); (**b**): OR with 95% confidence intervals (CI) against VOC (fully vaccinated BNT162b2); (**c**): OR with 95% confidence intervals (CI) against VOC (partially vaccinated mRNA-1273); (**d**): OR with 95% confidence intervals (CI) against VOC (fully vaccinated mRNA-1273).

**Figure 4 vaccines-10-00469-f004:**
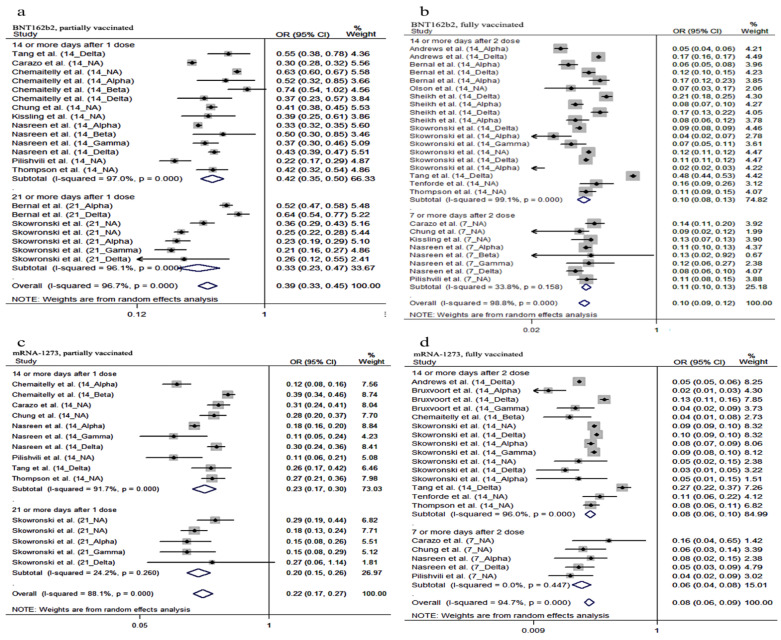
Forest plot odds ratio (OR) against COVID-19 by time (**a**): OR with 95% confidence intervals (CI) against COVID-19 infection of 14 or more days after being partially vaccinated (BNT162b2); (**b**): OR with 95% confidence intervals (CI) against COVID-19 infection of 14 or more days after being fully vaccinated (BNT162b2); (**c**): OR with its 95% confidence intervals (CI) against COVID-19 infection of 14 or more days after being partially vaccinated (mRNA-1273); (**d**): OR with its 95% confidence intervals (CI) against COVID-19 infection of 14 or more days after being fully vaccinated (mRNA-1273).

**Table 1 vaccines-10-00469-t001:** Main characteristics of the included articles.

Study/Country	Study Period	Study Population	Number of Cases/Controls	Age (Years)	Female (%)	Vaccine Type	Point Time of Assessing VE		Variants	NOS
							Days after 1 Dose	Days after 2 Dose		
Tang et al./Qatar [17]	23 March 2021 – 7 July 2021	Resident population	12,357/50,616	0–70+	21,095 (33.4)	BNT162b2, mRNA-1273	≥14	≥14	Delta	8
Skowronski et al./Canada [18]	4 April 2021 – 2 October 2021	Adults in British Columbia	3040/49,477	50–69	27,877 (53)	BNT162b2, mRNA-1273	≥21	—	NA Alpha, Gamma, Delta	6
Chung et al./Canada [19]	14 December 2020 – 19 April 2021	Community dwelling people who had symptoms of COVID-19	53,270/270,763	≥18	185,539 (57.3)	BNT162b2, mRNA-1273	≥14	≥7	NA	8
Bernal et al./UK [20]	October 2020 – May 2021	Symptomatic persons who underwent COVID-19 testing in England	11,356/96,371	≥16	NA	BNT162b2	≥21	≥14	Alpha, Delta	8
Thompson et al./US [21]	1 January 2021 – 22 June 2021	Laboratory-confirmed SARS-CoV-2 and COVID-19-like illness	3251/18,271	≥50	NA	BNT162b2, mRNA-1273	≥14	≥14	NA	8
Nasreen et al./Canada [22]	14 December 2020 – 3 August 2021	Symptomatic community-dwelling individuals	82,880/599,591	≥16	376,607 (55)	BNT162b2, mRNA-1273	≥14	≥7	Alpha, Beta, Gamma, Delta	7
Pilishvili et al./US [23]	28 December 2020 – 19 May 2021	Health care personnel across 25 U.S. states	1482/3449	≥18	4107 (83)	BNT162b2, mRNA-1273	≥14	≥7	Delta	6
Bruxvoort et al./US [24]	1 March 2021 – 27 September 2021	KPSC^a^ membership	2027/10,135	≥18	5364 (44.1)	mRNA-1273	—	≥14	Alpha, Gamma, Delta	7
Olson et al./US [25]	1 June2021 – 30 September 2021	Adolescent patients	179/285	12–18	210 (45.3)	BNT162b2	—	≥14	NA	6
Tenforde et al./US [26]	11 March 2021 – 5 May 2021	US adults	590/620	≥18	613 (50.7)	BNT162b2, mRNA-1273	—	≥14	NA	6
Bernal et al./UK [27]	8 December 2020 – 19 February 2021	Adults who reported symptoms	44,590/112,340	≥70	87,066 (55.5)	BNT162b2	—	≥14	Alpha	7
Chemaitelly et al./Qatar [28]	1 February 2021 – 10 May 2021	Resident population	66,042/66,042	19–39	40,298 (30.5)	mRNA-1273	≥14	≥14	Alpha, Beta	8
Sheikh et al./UK [29]	1 April 2021 – 6 June 2021	Individuals who have a PCR test for SARS-CoV-2 in study period	10,827/195,000	0–90+	NA	BNT162b2	—	≥14	Alpha, Delta	8
Carazo et al./Canada [30]	17 January 2021 – 5 June 2021	Healthcare workers	5316/53,160	18–74	48,513 (83)	BNT162b2, mRNA-1273	≥14	≥7	NA	6
Skowronski et al./Canada [31]	4 April 2021 – 1 May 2021	Community-dwelling adults in British Columbia	1226/15,767	≥70	8657 (50.9)	BNT162b2, mRNA-1273	≥21	—	NA	6
Skowronski et al./Canada [32]	3 May 2021 – 2 October 2021	Community-dwelling adults in British Columbia and Quebec	British:27,439/353,093 Quebec:17,234/837,681	≥18	British:211,999 (56) Quebee:511,620 (60)	BNT162b2, mRNA-1273	—	≥14	NA, Alpha, Gamma, Delta	7
Kissling et al. [33]	10 December 2021 – 31 May 2021	Adults in 8 European countries	592/4372	≥65	2933 (60)	BNT162b2	≥14	≥7	NA	6
Andrews et al./UK [34]	8 December 2020 – 3 September 2021	Symptomatic adults	1475,391/3757,981	≥16	2,928,972 (56)	BNT162b2, mRNA-1273	—	≥14	Alpha, Delta	8
Chemaitelly et al./Qatar [35]	1 January 2021 – 5 September 2021	Resident population	142,300/848,240	0–70+	NA	BNT162b2	≥21	—	NA Alpha, Beta, Delta	8

a. KPSC: integrated healthcare system with 15 hospitals and associated medical offices across Southern California.

**Table 2 vaccines-10-00469-t002:** Effectiveness of the BNT162b2 vaccine against SARS-CoV-2 infection after 1 dose and after 2 doses.

First Author	BNT162b2 OR ^1^				Variant
	14 Days after 1 Dose	21 Days after 1 Dose	7 Days after 2 Doses	14 Days after 2 Doses	
Chung et al. [19]	0.41 (0.38–0.45)	—	0.09 (0.02–0.12)	NA	NA
Bernal et al. [20]	—	0.525 (0.472–0.584)	—	0.063 (0.047–0.084)	Alpha
	—	0.644 (0.536–0.773)	—	0.12 (0.099–0.147)	Delta
Pilishvili et al. [23]	0.161 (0.121–0.213)	—	0.085 (0.06–0.121)	—	Delta
Thompson et al. [21]	0.42 (0.32–0.54)	—	—	0.11 (0.09–0.15)	NA
Tenforde et al. [26]	—	—	—	0.155 (0.091–0.262)	NA
Bernal et al. [27]	—	—	—	0.17 (0.12–0.23)	Alpha
Sheikh et al. [29]	—	—	—	0.21 (0.18–0.25)	Delta
	—	—	—	0.08 (0.07–0.10)	Alpha
	—	—	—	0.17 (0.13–0.22)	Delta
	—	—	—	0.08 (0.06–0.12)	Alpha
Chemaitelly et al. [35]	0.632 (0.598–0.668)	—	—	—	NA
	0.521 (0.321–0.845)	—	—	—	Alpha
	0.742 (0.539–1.02)	—	—	—	Beta
	0.366 (0.234–0574)	—	—	—	Delta
Skowronski et al. [31]	NA	0.36 (0.29–0.43)	—	—	NA
Nasreen et al. [22]	0.33 (0.32–0.35)	—	0.11 (0.10–0.13)	—	Alpha
	0.5 (0.30–0.85)	—	0.13 (0.02–0.92)	—	Beta
	0.37 (0.30–0.46)	—	0.12 (0.06–0.27)	—	Gamma
	0.43 (0.39–0.47)	—	0.08 (0.06–0.10)	—	Delta
Skowronski et al. [32]	—	—	—	0.09 (0.08–0.09)	Delta
	—	—	—	0.04 (0.02–0.07)	Alpha
	—	—	—	0.07 (0.05–0.11)	Gamma
	—	—	—	0.12 (0.11–0.12)	NA
	—	—	—	0.11 (0.11–0.12)	Delta
	—	—	—	0.02 (0.02–0.03)	Alpha
Carazo et al. [30]	0.297 (0.276–0.319)	—	0.145 (0.107–0.196)	—	NA
Kissling et al. [33]	0.39 (0.25–0.61)	—	0.13 (0.07–0.13)	—	NA
Skowronski et al. [18]	NA	0.25 (0.22–0.28)	—	—	NA
	NA	0.23 (0.19–0.29)	—	—	Alpha
	NA	0.21 (0.16–0.27)	—	—	Gamma
	NA	0.26 (0.12–0.55)	—	—	Delta
Tang et al. [17]	0.547 (0.384–0.78)	—	—	0.481 (0.436–0.53)	Delta
Andrews et al. [34]	—	—	—	0.05 (0.041–0.062)	Alpha
	—	—	—	0.165 (0.164–0.167)	Delta
Olson et al. [25]	—	—	—	0.07 (0.03–0.17)	NA

OR **^1^**: odds ratio (OR) against COVID-19 infection confirmed by PCR test.

**Table 3 vaccines-10-00469-t003:** Effectiveness of the mRNA-1273 vaccine against SARS-CoV-2 infection after 1 dose and after 2 doses.

First Author	mRNA-1273 OR ^1^				Variants
	14 Days after 1 Dose	21 Days after 1 Dose	7 Days after 2 Doses	14 Days after 2 Doses	
Chung et al. [19]	0.28 (0.20–0.37)	—	0.06 (0.03–0.14)	—	NA
Pilishvili et al. [23]	0.163 (0.124–0.214)	—	0.08 (0.057–0.113)	—	Delta
Thompson et al. [21]	0.27 (0.21–0.36)	—	—	0.08 (0.06–0.11)	NA
Tenforde et al. [26]	—	—	—	0.113 (0.059–0.215)	NA
Skowronski et al. [31]	—	0.29 (0.19–0.44)	—	—	NA
Nasreen et al. [22]	0.18 (0.16–0.20)	—	0.08 (0.02–0.15)	—	Alpha
	0.11 (0.05–0.24)	—	—	—	Gamma
	0.30 (0.24–0.36)	—	0.05 (0.03–0.09)	—	Delta
Skowronski et al. [32]	—	—	—	0.09 (0.09–0.10)	NA
	—	—	—	0.10 (0.09–0.10)	Delta
	—	—	—	0.08 (0.07–0.09)	Alpha
	—	—	—	0.09 (0.08–0.10)	Gamma
	—	—	—	0.05 (0.02–0.15)	NA
	—	—	—	0.03 (0.01–0.05)	Delta
	—	—	—	0.05 (0.01–0.15)	Alpha
Carazo et al. [30]	0.313 (0.241–0.405)	—	0.159 (0.039–0.651)	—	NA
Bruxvoort et al. [24]	—	—	—	0.016 (0.009–0.031)	Alpha
	—	—	—	0.133 (0.113–0.157)	Delta
	—	—	—	0.045 (0.022–0.091)	Gamma
Skowronski et al. [18]	—	0.18 (0.13–0.24)	—	—	NA
	—	0.15 (0.08–0.26)	—	—	Alpha
	—	0.15 (0.08–0.29)	—	—	Gamma
	—	0.27 (0.06–1.14)	—	—	Delta
Tang et al. [17]	0.263 (0.165–0.419)	—	—	0.269 (0.222–0.325)	Delta
Andrews et al. [34]	—	—	—	0.052 (0.048–0.056)	Delta
Chemaitelly et al. [28]	0.119 (0.085–0.163)	—	—	—	Alpha
	0.387 (0.345–0.455)	—	—	0.036 (0.013–0.081)	Beta

OR **^1^**: odds ratio (OR) against COVID-19 infection confirmed by PCR test.

## Data Availability

Not applicable.

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
