# Peer review of "Effectiveness of BNT162b2 and mRNA-1273 Vaccines against COVID-19 Infection: A Meta-Analysis of Test-Negative Design Studies"

_vaccines, 2022, doi:10.3390/vaccines10030469_

Round 1

Reviewer 1 Report

This paper describes the results of a meta-analysis of 19 studies that used a test negative design to assess vaccine effectiveness of the two COVID-19 mRNA vaccines. The methodology as described is good and the paper provides important information about how each vaccine performed under real world conditions with different variants.

Some revisions are needed. Table 1 needs to be reformatted. The number of days post-dose for each vaccine type are running together and making it hard to view the data. Also, please put the citation number by the study authors in Table 1 and in Figures 2, 3, and 4. I'm confused by the statement on page 4, lines 127 and 128 that says "numbers of literatures that reported the VE of BNT162b2 against infection was 18, mRNA-127 1273 12, and both BNT162b2 and mRNA-1273 11" since there are 19 studies included in the analysis. Please clarify how you are arriving at those numbers.

On page 10, lines 217 to 219, the authors account for the lower vaccine effectiveness in their study compared to in the clinical trials by saying, "In addition, some studies have reported that vaccines are less effective 217 in protecting older adults against COVID-19 infection, and three articles included in our 218 analysis studied people older than 50, which could reduce the overall vaccine protection". However, it is not clear why this would be the case as the clinical trials included participants who were over the age of 50 as well; please clarify that point.

Author Response

Thank you very much for your kind comments and suggestions. Please see the attachment.

Reviewer 2 Report

This is an interesting manuscript covering two COVID-19 vaccines effectiveness against COVID-19 infection. The manuscript underlines what is currently know and as such important data review for COVID-19 vaccines.  The analysis does not analyze the efficacy of long periods after vaccination and protection efficiency against severe illness and death in hospital, but still significant. The analyses contribute significantly to the evidence base and highlights the spots that may exist in vaccines effectiveness across type of vaccine, status of vaccination, and against variants of concerns. As SARS-COV-2 variants are emerging now and then, data of vaccines effectiveness are important especially with partially or fully vaccinated, and how often and timelines should individuals receive booster vaccine

Author Response

(The authors gave the same response as above.)

Reviewer 3 Report

Chang and colleagues studies the effectiveness of BNT162b2 and mRNA-1273 vaccines against COVID-19 infection using test negative design. They did the analysis on 19 articles, they found that the vaccine effectiveness (VE) against COVID19 is 90% and 92% for both vaccine, respectively in full vaccinated people. The VE is 85% and 91%, respectively  for the fully vaccinated group aganist Delta virses. While in partially vaccinated individuals, the VE  of BNT162b2 and mRNA-1273 was 61% and 22
78% against COVID-19 infection and was 53% and 71%, respectively aganist Delta virus infection.

In general it is a good review, however, I have some comments

1- The quality and arrangment of thr table is poor, this will cause confusion for the readers. The authors need to redesign/ rearrange  the table.

2- Some points need to be mentioned. The authors focused on VE, but I guess including the particptant ages, gender, risk factors, mortality rate, hospitilization is very important especially in cases of vaccibe faillure. This also will help to see if these points are comparable between COVID19 and Delta infections.

Author Response

(The authors gave the same response as above.)

Round 2

Reviewer 3 Report

No further suggestions